# MEMORIZATION FOR GOOD:
# ENCRYPTION WITH AUTOREGRESSIVE LANGUAGE MODELS

## ABSTRACT

Over-parameterized neural language models (LMs) can memorize and recite long sequences of training data. While such memorization is normally associated with undesired properties such as overfitting and information leaking, our work casts memorization as an unexplored *capability* of LMs. We propose the first symmetric encryption algorithm with autoregressive language models (SELM). We show that autoregressive LMs can encode arbitrary data into a compact real-valued vector (i.e., encryption) and then losslessly decode the vector to the original message (i.e., decryption) via random subspace optimization and greedy decoding. While SELM is not amenable to conventional cryptanalysis, we investigate its security through a novel empirical variant of the classic IND-CPA (indistinguishability under chosen-plaintext attack) game and show promising results on security.[1]

## 1 INTRODUCTION

Pre-trained language models (LMs) (Devlin et al., 2018; Brown et al., 2020) are the foundation of virtually all state-of-the-art natural language processing methods. Generalization is central to LMs' success; i.e., they can assign plausible probabilities to unseen token sequences. Counter to generalization is *memorization*, when an LM assigns abnormally high probabilities to token sequences seen during training. Existing work raises concerns about memorization because it compromises language generation quality Lee et al. (2022) and can reveal private training data (Carlini et al., 2021). Unintended memorization is thus generally considered a weakness of LMs (Bommasani et al., 2021).

In parallel, contemporary symmetric encryption algorithms are based on two different structures: Substitution Permutation Networks (SPN) or Feistel Networks (Feistel, 1973).[2] A symmetric encryption algorithm enables two parties (Alice and Bob) to communicate privately, preventing an eavesdropping third party (Eve) from reading their messages (also called plaintexts). Cryptanalyses (attacks on symmetric encryption algorithms) take advantage of the homogeneous structure across algorithms and often generalize across multiple encryption algorithms. For example, differential cryptanalysis (Biham & Shamir, 1993) was a new state-of-the-art attack on five separate encryption algorithms: FEAL (Feistel), Khafre (Feistel), REDOC II (SPN), LOKI89 (Feistel) and Lucifer (Feistel + SPN).

We frame LM memorization as an under-explored *skill* and develop **symmetric encryption with autoregressive language models** (SELM), the first symmetric encryption algorithm based on LM memorization. SELM diversifies the available portfolio of cryptography primitives; new cryptanalyses would likely not generalize from AES or DES to SELM because of SELM's novel structure.

In a naive formulation of SELM, Alice fine-tunes a public autoregressive LM with pre-trained parameters $\theta_0^D$ ($D$ is the LM's number of trainable parameters) until it memorizes her message; i.e., greedy decoding produces the message verbatim. She sends the change in parameters $\Delta\theta^D$, which is the *ciphertext*, to Bob, who applies the update to the same public LM (by simply adding $\Delta\theta^D$ to $\theta_0^D$) and runs greedy decoding to regenerate Alice's message. This process is lossless and is guaranteed to exactly generate the original message.

---

[1] Our code and data will be made publicly available.
[2] For example, AES and GOST are SPNs; DES and Twofish are Feistel Networks.

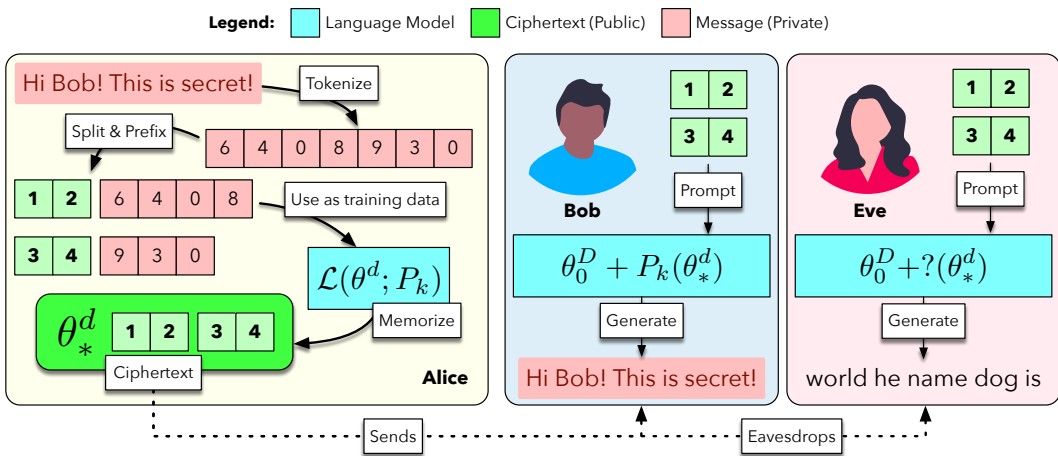

Figure 1: Alice and Bob use SELM to communicate privately despite Eve eavesdropping. **Left:** Alice encrypts her message and then sends the low-dimensional vector $\theta_*^d$ and her random prompts as the ciphertext $c$. **Middle:** Bob reconstructs Alice's fine-tuned LM by projecting $\theta_*^d$ onto $\mathbb{R}^D$ with the shared secret projection $P_k$ and regenerates Alice's message using the same prompts Alice sent. **Right:** Eve does not have the secret projection $P_k$ so Alice's random prompts generate gibberish.

However, this formulation has two immediate problems. First, the parameter update $\Delta\theta^D$ is the same size as the LM. For example, using GPT-2$_{small}$ (Radford et al., 2019) with 124M parameters would produce 496MB ciphertexts (with FP32) even for very short messages. Second, *Eve* can read Alice's message—she can also add $\Delta\theta^D$ to the public LM parameters $\theta_0^D$ to regenerate Alice's message; this encryption algorithm is *not secure*.

Both problems are independently solvable. Low-rank fine-tuning methods like Adapter (Houlsby et al., 2019), prefix tuning (Li & Liang, 2021), or BitFit (Ben Zaken et al., 2022) would reduce the ciphertext size. Alice would send only a change vector of the tuned parameters, which is typically a small fraction of $D$. To solve the security problem, we could use secret pre-trained language models unique to Alice and Bob as their shared *secret key*. Eve would lack the secret pre-trained weights needed to decode the original message. However, every pair of parties would need a separate pre-trained LM, which is computationally unfeasible.

Instead, SELM presents an elegant solution that solves both problems simultaneously (Fig. 1). Our solution is inspired by *intrinsic dimension* (Li et al., 2018, Section 2 contains more background). It aims to quantify a learning task's intrinsic difficulty through *random subspace optimization*, where a low-dimensional vector $\theta^d$ is projected onto the original $D$-dimensional parameter space via a random projection $P$: $\theta^D = P(\theta^d)$. Gradient descent minimizes the loss with respect to $\theta^d$ rather than $\theta^D$, while $P$ is frozen during optimization.

While Li et al. investigate a wholly different problem, random subspace optimization offers an appealing tool for encryption. First, it solves the size problem: we represent ciphertexts using the low-dimensional vector $\Delta\theta^d$ rather than $\Delta\theta^D$. It reduces ciphertext sizes more than $100\times$ and upwards of $100,000\times$ in our work. Second, we propose a security-motivated form of random subspace optimization: we parameterize the random projection $P$ by Alice and Bob's secret key $k$ so $P_k$ is a deterministic function of $k$. Such *secret* subspace optimization addresses the security problem: Eve cannot regenerate Alice's message because she doesn't know which subspace of $\mathbb{R}^D$ Alice and Bob are using. She cannot project $\theta^d$ onto the $D$-dimensional parameter space without the secret projection $P_k$ (see Fig. 2, right) and she cannot construct $P_k$ because she lacks the secret key $k$.

We formally describe SELM's encryption and decryption steps and investigate its properties, including what's encryptable and SELM's security. We establish that LMs optimized in random subspaces can achieve perfectly memorize both arbitrary data and long sequences of English text. Because SELM is the first exploration of neural LMs for encryption, it is not amenable to typical cryptanalysis for symmetric encryption algorithms. Instead, we investigate its security through a novel empirical variant of the classic IND-CPA (indistinguishability under chosen-plaintext attack) game. We expose security weaknesses through our empirical IND-CPA game and propose regularization strategies

to improve security. We present a novel application of LMs' text-in, text-out interface and their memorization capability; it exemplifies a rich, under-explored venue for further investigation.

## 2 BACKGROUND

**Random Subspace Optimization** Li et al. (2018) propose minimizing a loss function $\mathcal{L}$ with $D$ parameters $\theta^D$ in a random $d$-dimensional subspace to measure the intrinsic difficulty of a given optimization problem. If an approximate solution (i.e., 90% as good as the original solution) with as few as $d$ parameters exist, then the function's intrinsic dimension is $d$. Harder optimization problems have larger intrinsic dimensions. More formally, $\mathcal{L}$ is minimized with respect to:

$$\theta^D = \theta_0^D + P(\theta^d),\tag{1}$$

where $\theta_0^D$ is the initial parameter vector in the original $D$-dimensional parameter space and $P \colon \mathbb{R}^d \to \mathbb{R}^D$ is the Fastfood Transform, an algorithm to efficiently multiply by a Gaussian matrix (Le et al., 2013). Only $\theta^d$ is tunable during optimization; $P$ is frozen. For example, suppose $D = 3$ and $d = 2$. In Fig. 2, optimization starts at $\theta_0^D$ (the origin) and moves along the 2-dimensional subspace (defined by $P$) until it reaches a solution $\theta_*^d$ (and hence $\theta_*^D$).

**Cryptography** A symmetric encryption algorithm lets two parties, Alice and Bob, send private messages without a third party, Eve, learning anything about the messages. Symmetric encryption algorithms encrypt a message $m$ into a ciphertext $c$. The encryption of $m$ to $c$ is parameterized by a secret key $k$ known only by Alice and Bob. Without $k$, Eve should not be able to learn anything about $m$ from $c$. In Fig. 1, Alice sends Bob a secret message after encrypting it. Eve can read the ciphertext $c$ but she cannot read the *message* because she lacks the key $k$.

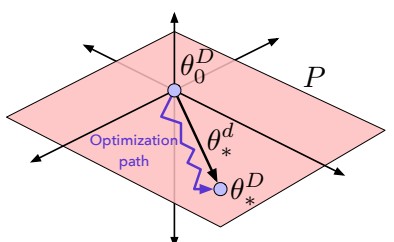

Figure 2: Visualization of random subspace optimization. The original model has 3 parameters $\theta_0^D$ and the lower-dimensional vector $\theta^d$ has 2 parameters. Optimization is limited to the 2-dimensional subspace fixed by $P$.

## 3 ALGORITHM

SELM is a symmetric cipher with an encryption algorithm $E$ and a decryption algorithm $D$. Given a key $k$, $E$ encrypts a message $m$ into a real-valued $d$-dimensional vector $\theta_*^d$, while $D$ decrypts the ciphertext $\theta_*^d$ back to the same message $m$.

We use a toy running example to illustrate how SELM works. Suppose Alice wants to send Bob the message "Hello Bob! This is secret!" They have a shared secret key $k = 1743$ and *Oscar*, a toy LM with publicly available weights $\theta_0^D$. SELM has the following steps for encryption $E$ (visualized in Fig. 1, left). Alice must:

1. Tokenize the message $m$. Oscar tokenizes "Hello Bob! This is secret" to a sequence of token IDs 6, 4, 0, 8, 9, 3, 0 .[3]
2. Convert $m$'s tokens into training examples, prefixed with a unique prompt, which speed up memorization (see Algorithm). Oscar's maximum length is 6 tokens, so Alice splits her 7-token message into two examples.[4] Alice randomly chooses **1, 2** and **3, 4** as prompts, so 6, 4, 0, 8, 9, 3, 0 turns to **1, 2** 6, 4, 0, 8 and **3, 4** 9, 3, 0 .
3. Generate a Fastfood projection $P_k$ via random sampling (exactly as in Le et al. (2013)) after setting the key $k = 1743$ as the random seed. See Appendix A for a detailed review of Le et al..
4. Initialize the $d$-dimensional vector $\theta^d$ to the zero vector $\vec{0} \in \mathbb{R}^d$ so optimization starts at the LM's pre-trained parameters $\theta_0^D$.
5. Minimize cross-entropy over next-token-prediction over message tokens only with respect to $\theta^d$.
6. Stop training when greedy decoding, conditioned on each prompt, generates the correct message. Alice stops when her LM generates 6, 4, 0, 8 given **1, 2** and 9, 3, 0 given **3, 4** .
7. Output the $d$-dimensional vector $\theta_*^d$ and the prompts **1, 2** , **3, 4** as the ciphertext $c$.

---

[3]We use red to denote message tokens and **bold/green** to denote prompt tokens, as in Fig. 1.

[4]Real LMs have a maximum length of 1,024 or more.

For decryption $D$, Bob must:

1. Generate the Fastfood projection $P_k$ using the same process and key $k = 1743$ as Alice. Bob and Alice generate the same projection $P_k$.
2. Reconstruct Alice's fine-tuned LM by projecting $\theta_*^d$ onto $\mathbb{R}^D$ via $P_k$: $\theta_*^D = \theta_0^D + P_k(\theta_*^d)$.
3. Prompt the LM parameterized by $\theta_*^D$ to generate the messages. Bob prompts with 1, 2 and 3, 4 and generates 6, 4, 0, 8 and 9, 3, 0, respectively.
4. Join and de-tokenize the tokens. Oscar de-tokenizes 6, 4, 0, 8, 9, 3, 0 to "Hi Bob! This is secret!"

Intuitively, secret subspace optimization simultaneously addresses the size and security issues. First, $\theta_*^d$ is significantly smaller and easier to share than $\theta_*^D$ because $d \ll D$. $d$ is 700 to 160,000 times smaller than $D$ in our experiments. Second, Eve cannot recover the private message $m$ from $\theta_*^d$ because she cannot project $\theta_*^d$ onto the **public** pre-trained parameters $\theta_0^D$ without $P_k$. For example, in Fig. 2, Eve intercepts the 2-dimensional vector $\theta_*^d$, but without the projection $P_k$, she cannot orient $\theta_*^d$ in 3-dimensional space to find the true parameters $\theta_*^D$, despite knowing $\theta_0^D$. We further review security in Section 5.

**Implementation Details** The LM tokenizer must be invertible: it cannot have any `[UNK]` tokens, because they cannot be mapped back to the original message. Note that many popular autoregressive LMs, including GPT-2, GPT-3 (Brown et al., 2020), LLama (Touvron et al., 2023a) and LLama 2 (Touvron et al., 2023b) have invertible tokenizers.

While an LM is capable of memorizing messages directly, prefixing a message with a random prompt that the LM is unlikely to have seen in pre-training helps reset the biases learned in pre-training, e.g., a sentence often starts with an article like "the" or "a". Such biases may make memorizing some messages that do not conform to these biases harder because the LM needs to first *unlearn* the biases. A random prompt softly serves as a fresh start for the LM and makes memorization easier (Carlini et al., 2022; Tirumala et al., 2022). We measure the effect of different prompts and prompt lengths on memorization speed and find that random UUIDs (universally unique identifiers) are empirically the best prompts; the same UUIDs likely do not appear in the pre-training corpus. We use UUIDs for our remaining experiments. Full results for our prompt experiments are in Appendix B.

We omit a minor step in SELM's description. We use a standard, generic hybrid construction to turn SELM into a probabilistic cipher (Boneh & Shoup, 2020, Section 5.4.1), briefly explained here. Before generating $P_k$ (Encryption, Step 3), Alice actually randomly samples an integer $x$ and seeds a pseudo-random function $F$ with $k$ and $x$ to generate a new, random key $k'$. She generates $P_k$ with $k'$ and sends $x$ in the ciphertext. Bob uses $k$ and $x$ to generate $k'$, which generates $P_k$ (Decryption, Step 1). Without this step, every message would use the same secret subspace, enabling Eve to learn a mapping from $\mathbb{R}^d$ to plaintext messages. See Appendix E for more details.

## 4 WHAT CAN BE ENCRYPTED?

In this section, we explore whether an LM with random subspace optimization can memorize data, and if it can, *what* data it can memorize. Complementary to that goal, we want to understand what factors affect an LM's memorization speed (i.e., number of epochs until perfect memorization). We empirically show that LMs can memorize completely random noise, even when optimized in low-dimensional subspaces. **Memorizing random noise leads us to conclude that SELM can encrypt arbitrary messages**, given sufficient time (number of epochs) and free parameters (dimension of $\theta^d$).[5] To the best of our knowledge, this is the first systematic study on the memorization capability of LMs on arbitrary data and random subspaces.

### 4.1 EXPERIMENTAL SETUP

Our exploration is based on GPT-2$_{\text{small}}$ (Radford et al., 2019, 124M parameters, uncased). To measure its memorization cability, we fine-tune it in $d$-dimensional subspaces (as described in Section 3).

By default, we encrypt news articles from the XSum dataset (Narayan et al., 2018) truncated at 100 tokens because they are similar to (but not included in) GPT-2's pre-training data. We vary message length, data domain and the underlying LM to measure their effects on memorization speed. We

---

[5]See Appendix D.1 for more discussion.

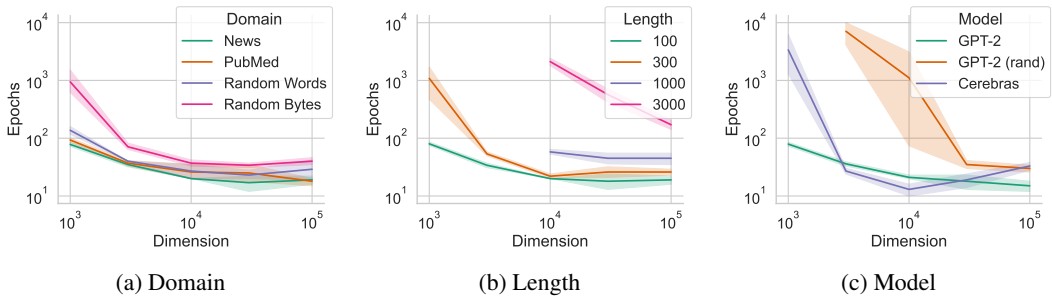

(a) Domain        (b) Length        (c) Model

Figure 3: Effects of message source, message length, and LM on memorization speed (number of epochs until perfectly memorized). Missing values indicate failures to perfectly memorize in 10K epochs (note that SELM can encrypt anything; see Appendix D.2). Shaded bars indicate 95% confidence intervals over 10 trials. GPT-2 (rand) is not pre-trained; it is randomly initialized.

vary message length by truncating news articles at 100, 300, 1,000 and 3,000 tokens. We vary data domain by sampling from three other domains, all truncated at 100 tokens:

1. **PubMed**: We use PubMed abstracts because they are written in domain-specific English which is different from GPT-2's pre-training domain.[6]
2. **Random Words**: We generate a set of unique words from 10K Wikipedia articles using NLTK tokenization (Bird & Loper, 2004), then randomly sample words and join them with spaces until we reach the token limit. We expect the lack of linguistic structure to slow down memorization.
3. **Random Bytes**: We uniformly sample random bytes (integers in $[0, 255]$) until we reach the token limit. This is simply uniform random noise (the highest entropy distribution).

We sample 10 messages for every domain and length. We evaluate how pre-training affects memorization speed by comparing GPT-2$_{small}$ (124M parameters), a randomly initialized GPT-2$_{small}$ (124M parameters) and Cerebras's 111M (Dey et al., 2023) models, trained on the Pile (Gao et al., 2020).

In all of our experiments, we use $d = $ 1K, 3K, 10K, 30K and 100K, stopping experiments after 10,000 epochs because of compute restraints.[7] As far as we know, there is no existing work on LMs memorizing data in a random subspace. Thus, there are no obvious default hyperparameters. After a broad hyperparameter search, we linearly decay learning rate over 2,000 epochs, clip gradients with L2 norms above $10^5$ and disable all dropout and weight decay. A complete discussion of our hyperparameter choices is in Appendix C.

### 4.2 DISCUSSION

**First**, we observe that LMs optimized in a random subspace can perfectly memorize *entirely random data*. In Fig. 3a, GPT-2 with 1,000 free parameters memorizes random orderings of bytes in under 1,000 epochs. The constrained model perfectly memorizes data drawn from the highest-possible entropy distribution. As far we know, this is the first demonstration of such a capability of LMs. **Second**, messages that are closer to the LM's pre-training data (News, followed by PubMed and then random words) are easier to memorize, as expected (see Appendix D.3 for more discussion on this). **Third**, adding more intrinsic dimension parameters only speeds up *difficult* memorization. In Fig. 3b, GPT-2 with 10K free parameters memorizes 3,000-token messages in 1,673 epochs. Adding 90K free parameters (for a total of 100K) speeds up memorization by 89% on average (1,673 to 184 epochs). In contrast, 1,000-token messages do not become much easier to memorize with more than 10K parameters: 100K parameters only speeds up memorization by 22% on average (58 to 45 epochs). **Finally**, different LMs are stronger or weaker memorizers. Pre-training improves memorization: the randomly initialized model cannot efficiently memorize messages with only 1K free parameters, while a pre-trained GPT-2$_{small}$ can. Cerebras's 110M uses hyperparameters tuned for GPT-2$_{small}$ and is competitive at $d \geq 3000$, demonstrating that SELM is LM-agnostic.

---

[6] https://pubmed.ncbi.nlm.nih.gov/

[7] "Epoch" refers to a full pass over all the training examples; a 3000-token message has four training examples. On an A6000 GPU, 100 epochs with GPT-2$_{small}$ takes about 1.5 minutes.

### 4.3 SPEED & SIZE TRADE-OFF

A smaller $d$ leads to a smaller ciphertext, but also comes at a cost—encryption takes longer. SELM's ciphertext sizes depend only on $d$, not on the input length; messages of different lengths are always encoded into a $d$-dimensional vector. Other symmetric ciphers like the Data Encryption Standard (National Bureau of Standards, 1977, DES) and the Advanced Encryption Standard (Pub, 1999, AES) do not have this flexibility; the ciphertext is always the same size as the message.

This hyperparameter $d$ leads to a trade-off: encrypting long sequences of data (e.g., movies) in a time-sensitive application requires a larger $d$ because longer non-text messages take longer to memorize (see Fig. 3). Encrypting many short messages can use a smaller $d$ because it is more space-efficient and short messages are easy to memorize, even with few free parameters.

## 5 SECURITY

Symmetric ciphers should stop Eve from learning anything about the message from the ciphertext. Secret subspace optimization intuitively prevents Eve from decrypting a ciphertext $c$ into its message $m$ (see Fig. 2). Unfortunately, it's hard to measure if a ciphertext $c$ reveals *any* information about its message $m$ to Eve. Goldwasser & Micali (1984) proved that Alice and Bob winning the IND-CPA security game means Eve cannot learn any information about the message from its ciphertext. We review the IND-CPA game, describe our empirical variant, and discuss our experimental results.

### 5.1 IND-CPA GAME

The IND-CPA game (indistinguishability under chosen-plaintext attack, see Bellare & Rogaway, 2005) quantifies Eve's ability to distinguish which message produced a ciphertext (which measures how much information Eve learns from a ciphertext). It follows these steps:

1. Eve sends a message $m$ to Alice.
2. Alice encrypts the message into a ciphertext $c$ and returns it to Eve.
3. Eve and Alice repeat Steps 1 and 2 as many times as Eve likes.
4. Eve sends two messages $m_0$ and $m_1$ with the same length to Alice.
5. Alice encrypts one randomly chosen message $m_i$ into a ciphertext $c$ and returns it to Eve.
6. Eve looks at $c$ and guesses which message, $m_0$ or $m_1$, was encrypted.

If the encryption is perfectly secure, Eve cannot learn *anything* from the ciphertext $c$, so she can only guess correctly 50% of the time. If Eve guesses correctly *more* than 50% of the time, she must be learning *something* from the ciphertext. Thus, if Eve cannot win the game, she cannot decrypt ciphertexts in any capacity (conversely, decrypting ciphertexts trivially wins the IND-CPA game).

We re-frame the IND-CPA game as a binary classification problem:

1. Choose messages $m_0$ and $m_1$.
2. Encrypt them many times to form a training set of $t$ examples: $\{(c_i, m_i)\}_{i=1}^{t}$.
3. Train a binary classification model to predict $m_i$ from $c_i$.
4. Test the models on a held-out set of $s$ examples: $\{(c_i, m_i)\}_{i=1}^{s}$.

If a model is correct more than 50% of the time, it must be learning something from the ciphertext $c$. We model the classification models' success rates as binomial distributions and evaluate the null hypthosis that a model is random ($p = \frac{1}{2}$) using a binomial test. We reject the null hypothesis in favor of the alternative hypothesis that a model is stronger than random ($p > \frac{1}{2}$) for p-values less than 0.05.

**Why a modified security game over a proof or contemporary cryptanalysis?** Symmetric encryption algorithms rarely, if ever, depend on existing hard problems like prime factoring.[8] Popular algorithms like AES (Dworkin et al., 2001) and newer algorithms like SPECK 32/64 (Beaulieu et al., 2015) and Grain-128AEAD (Hell et al., 2021) are not provably "hard"; instead they resist the strongest published cryptanalysis techniques.

Unfortunately, contemporary cryptanalysis techniques make assumptions about the analyzed algorithm. For example, the avalanche test (Webster & Tavares, 1985) assumes the cipher operates

---

[8]In contrast, public key encryption systems like RSA (Rivest et al., 1978) are provably as hard as prime factoring mod $n$, which has no polynomial-time algorithms.

Table 1: Test accuracies from the empirical IND-CPA game described in Section 5.1. $\theta^d$ columns use the full 10,000-dimensional ciphertext as input; $f(\theta^d)$ columns use the 6-dimensional feature vector from Section 5.2. We evaluate if a model is randomly guessing using a binomial test. **Bolded** numbers indicate we **reject** the null hypothesis that a model is randomly guessing with $p < 0.05$, implying that the model learns something from the ciphertext.

| Algorithm | $m_1$ | KNN | | LDA | | SVM | | GradBoost | | FFNN | |
|---|---|---|---|---|---|---|---|---|---|---|---|
| | | $\theta^d$ | $f(\theta^d)$ | $\theta^d$ | $f(\theta^d)$ | $\theta^d$ | $f(\theta^d)$ | $\theta^d$ | $f(\theta^d)$ | $\theta^d$ | $f(\theta^d)$ |
| Original | News (N1) | 0.50 | **0.77** | 0.44 | **0.76** | **0.78** | **0.77** | 0.52 | **0.75** | 0.48 | **0.77** |
| | PubMed (PM) | 0.50 | **0.59** | 0.49 | **0.84** | **0.81** | **0.87** | 0.54 | **0.82** | **0.57** | **0.86** |
| | Rand. Words (RW) | 0.50 | **0.69** | 0.54 | **0.85** | **0.73** | **0.86** | 0.45 | **0.82** | 0.50 | **0.83** |
| | Rand. Bytes (RB) | 0.50 | **1.00** | **0.58** | **1.00** | **1.00** | **1.00** | **0.70** | **1.00** | **1.00** | **1.00** |
| L2 Reg. | News (N1) | 0.54 | 0.49 | 0.48 | **0.60** | 0.54 | **0.59** | 0.53 | **0.58** | 0.51 | **0.61** |
| | PubMed (PM) | 0.48 | 0.41 | **0.58** | **0.55** | 0.50 | 0.48 | 0.49 | 0.47 | 0.49 | 0.52 |
| | Rand. Words (RW) | 0.50 | **0.79** | 0.48 | **0.85** | **0.65** | **0.84** | 0.55 | **0.83** | 0.51 | **0.75** |
| | Rand. Bytes (RB) | 0.49 | **0.92** | 0.55 | **0.99** | **0.79** | **0.99** | 0.48 | **0.99** | 0.51 | **0.87** |
| Dist. Reg. | News (N1) | 0.47 | 0.46 | 0.49 | 0.49 | 0.48 | 0.52 | 0.46 | 0.51 | 0.48 | 0.46 |
| | PubMed (PM) | 0.55 | 0.50 | 0.49 | 0.49 | 0.54 | 0.49 | 0.54 | 0.47 | 0.54 | 0.41 |
| | Rand. Words (RW) | 0.47 | 0.48 | 0.55 | 0.47 | 0.49 | 0.49 | 0.45 | 0.44 | 0.47 | 0.47 |
| | Rand. Bytes (RB) | 0.50 | 0.47 | 0.52 | 0.45 | 0.50 | 0.47 | 0.48 | **0.58** | 0.49 | 0.55 |

on bit-sequences. Differential cryptanalysis (Biham & Shamir, 1993) and its variants assumes the cipher operates on bit-sequences and the cipher is a substitution permutation network or Feistel network. Our neural cipher does not satisfy any of these assumptions; the IND-CPA game makes no assumptions about the cipher and is therefore suitable for our proposed algorithm. We explain further in Appendix E.2.

## 5.2 EXPERIMENTAL SETUP

We aim to rigorously test SELM's security to highlight and patch any weaknesses. We play our empirical IND-CPA game with four $\langle m_0, m_1 \rangle$ pairs and five binary classification models. $m_0$ is a message from the news domain (referred to as N0). We use a randomly sampled message from each domain (news, PubMed, random words, random bytes, referred to as N1, PM, RW and RB, respectively) for $m_1$ (Step 1 in Section 5.1). To test SELM's security in the worst-case scenario, we make it as easy as possible for Eve; we hypothesize that different message domains will have different ciphertext distributions in $\mathbb{R}^d$, facilitating classification.

We use 100-token messages and $d = 10K$ because we need to encrypt each example many times and shorter messages with larger $d$ are faster to encrypt (see Section 4.2). We encrypt 500 examples of each message (400 training examples, 100 test examples) with GPT-2$_{small}$. We pair $m_0$ with each $m_1$ to play four instances of the IND-CPA game with 800/200 train/test examples, respectively.

We train five different binary classification models: (1) a K-nearest neighbors model (Cover & Hart, 1967, KNN) and (2) a linear discriminant analysis (LDA) model representing linear classifiers, (3) a support vector machine (Cortes & Vapnik, 1995, SVM) to model non-linear interactions between the input features, (4) gradient-Boosted decision trees (Friedman, 2001, GradBoost) as a strong binary classification model that rarely overfits, and (5) A two-layer feed-forward neural network with ReLU non-linearity (Agarap, 2018, FFNN). Appendix F contains specific details for each model. At test time, each model, given a ciphertext, predicts which message produced it. Complementary to our main results using the ciphertext as input, we also test each model using a hand-crafted feature function $f : \mathbb{R}^d \to \mathbb{R}^6$ with six features: the mean, standard deviation, maximum, minimum, L1 norm and L2 norm of the values in the ciphertext.

## 5.3 RESULTS

Our original algorithm, while intuitively secure, loses the IND-CPA game to all four models (see Table 1). To better understand the exploitable patterns in ciphertext distributions, we visualize them (for N0 and RB) with T-SNE in Fig. 4a. Ideally, the ciphertext distribution in $\mathbb{R}^d$ for different

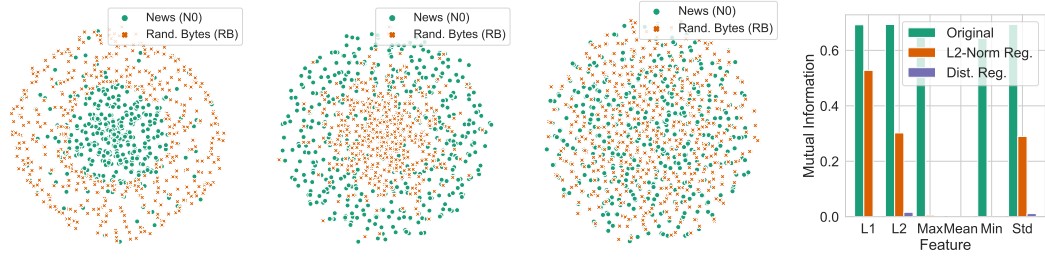

(a) Original algorithm    (b) L2 norm regularization    (c) Dist. regularization    (d) Mutual information

Figure 4: T-SNE (Van der Maaten & Hinton, 2008) visualizations of the N0 and RB ciphertexts for each algorithm variant. We limit the visualization to two messages that are reliably distinguished for visual clarity. **(a)** RB ciphertexts are more spread out in $\mathbb{R}^d$ than N0 ciphertexts. **(b)** After L2-norm regularization, RB ciphertexts have smaller L2 norms than N0 ciphertexts. **(c)** Distribution normalization evenly spreads the ciphertexts from RB and N0 in $\mathbb{R}^d$. **(d)** Mutual information estimations on the training set for each ciphertext feature.

messages should be identical. However, RB's ciphertexts are clearly more spread out than N0's ciphertexts. We use mutual information estimation on the ciphertext's features (Pedregosa et al., 2011) to quantify this spread's effect on binary classification. Fig. 4d shows that L2 norm, L1 norm and standard deviation are the strongest predictors for the binary classification models.

## 5.4 REGULARIZATION

To minimize differences in ciphertext distributions, we introduce L2 regularization during training that penalizes distance from a non-zero target L2 norm:

$$\mathcal{L}(\theta^d) = \sum_i p(t_i|t_1 \ldots t_{i-1}; \theta^d) + \lambda \left| ||\theta^d||_2 - \alpha \right|, \tag{2}$$

where the regularization coefficient $\lambda$ and the target L2 norm $\alpha$ are both hyperparameters. All ciphertexts $c$, no matter the message $m$ or key $k$, should satisfy $||c_i||_2 \approx \alpha$. For 100-token messages with $d = 10K$, we choose $\alpha = 2 \cdot 10^{-5}$ based on unregularized ciphertexts' L2 norms and a linear $\lambda$ schedule from 0 to $10^5$ over 500 epochs. $\lambda$ and its schedule are tuned to maximize $\lambda$ while still encrypting all messages (see Appendix C).

Simple regularization based on L2 norm may be insufficient to eliminate all patterns. We explore an alternative regularization term that penalizes differences between a ciphertext $c \in \mathbb{R}^d$ and a $d$-dimensional sample from a univariate normal distribution $\mathcal{N}(0, \sigma^2)$, where $\sigma$ is a hyperparameter. Formally, we minimize 1D Wasserstein distance between $\theta^d$ and a random $d$-dimensional sample drawn from a normal distribution $x \sim \mathcal{N}(0, \sigma^2) \in \mathbb{R}^d$:

$$\Omega(\theta^d; \sigma) \triangleq \int |\theta^{(d)} - x| dx, \tag{3}$$

$$\mathcal{L}(\theta^d) = \sum_i p(t_i|t_1 \ldots t_{i-1}; \theta^d) + \lambda \Omega(\theta^d; \sigma). \tag{4}$$

We choose a normal distribution because unregularized ciphertexts look like normal distribution samples (see Appendix G for examples and implementation details). We choose a fixed $\sigma$ for all messages: $\sigma = 4 \cdot 10^{-7}, \lambda = 5 \cdot 10^8$. This term encourages optimization to find ciphertexts that are indistinguishable by any feature. We repeat the IND-CPA game for each encryption algorithm variant. Table 1 contains the results.

## 5.5 REGULARIZATION RESULTS

Regularization significantly improves SELM's success rate at the IND-CPA game. Only gradient-boosted decision trees win the IND-CPA game against the distribution regularized variant, and their accuracy is only 8% above chance. Stronger regularization would further improve SELM's security, but we cannot increase $\lambda$ because it would prevent perfect memorization.

To understand the regularization's effects on ciphertext distributions, we visualize ciphertext distributions for our regularized variants in Fig. 4. T-SNE reliably separates L2-regularized N0 and RB ciphertexts, indicating that L2-regularization does not identically distribute ciphertexts in $\mathbb{R}^d$. T-SNE evenly spreads distribution-regularized ciphertexts, suggesting that distribution regularization does identically distribute ciphertexts in $\mathbb{R}^d$.

## 6 RELATED WORK

**Language Model Memorization** Pre-trained LMs memorize training data, even when it is only seen once during training. Carlini et al. (2019) frame memorization as a "persistent, hard-to-avoid issue." We re-frame memorization as a skill and propose an application of LM memorization. In fact, work that measures what "worsens" memorization can be re-framed as work that improves our encryption algorithm. Carlini et al. (2021) develop manual prompting techniques to extract training data from pre-trained LMs and raise the concern that it's easier to extract training data from larger models. Carlini et al. (2022) find three relationships that correlate with memorization: (1) model capacity, (2) number of repeated examples and (3) context length in tokens. While this is concerning for typical LM applications, these trends support future work in the vein we have proposed. Khandelwal et al. (2020) use explicit memorization to improve LM generalization on unseen data. SELM is also a positive application of memorization, but we directly exploit a LM memorization rather than storing data external to the LM's weights.

**Machine-Learning Cryptanalysis** Gohr (2019) proposes the first machine learning-based cryptographic distinguisher: a "residual tower of two-layer convolutional neural networks" trained to classify inputs as real differentials or random data. Wenger et al. (2022) propose transformer-based attacks on lattice cryptography algorithms. Our machine-learning-based cryptanalysis draws inspiration from these works but uses simpler models.

**Neural Cryptography** While machine learning has recently improved traditional cryptanalysis techniques, it is significantly more difficult to develop a cryptographic primitive based on machine learning, especially one as sophisticated as a symmetric cipher. Kanter et al. (2002) propose a novel key-exchange protocol based on the synchronization of two randomly initialized neural networks. Alice and Bob would use the protocol to securely exchange keys, while SELM encrypts data by assuming Alice and Bob already exchanged a key. The authors use simulation to analyze their protocol's security. Klimov et al. (2002) analyze the proposed protocol and develop three separate successful attacks, proving it insecure.

## 7 CONCLUSION & FUTURE WORK

We propose SELM, a novel symmetric cipher based on pre-trained LMs' exceptional ability to memorize data even constrained to a random subspace of its full parameter space. We find that LMs can memorize long sequences and even random noise, indicating that SELM can encrypt anything. We then adapt random subspace optimization to *secret* subspace optimization, where the random subspace becomes a deterministic function of a secret key $k$, and analyze SELM's security properties using an empirical variant of the traditional IND-CPA game.

LMs' inherent over-parameterization implies that empirically indistinguishable message representations should exist. While we propose regularization strategies that improve security, SELM is still not yet semantically secure. We hypothesize that current security weaknesses are due to GPT-2's memorization ability relative to the regularization strength: stronger memorization enables stronger regularization. Potential solutions include: (1) Larger language models: Carlini et al. (2021) find that larger models memorize more (2) Longer prompts (in tokens): Carlini et al. (2022) find that longer prompts cause more memorization. (3) Better regularization: our Wasserstein-based regularization requires sorting $\theta^d$, which likely harms optimization. Many CPUs today have specialized instructions to support AES (Akdemir et al., 2010) encryption speed. Similarly, better hardware support for autoregressive LMs will improve our algorithm's speed and viability.[9]

We hope our work inspires future explorations of both ML in cryptography and LM memorization as a strength instead of a weakness for use in novel LM applications.

---

[9]See Appendix H for further discussion of our work's limitations.

## REPRODUCIBILITY STATEMENT

Because of the technical novelty, we make the code available as supplementary material. The code also includes scripts and instructions to generate all figures (Figs. 3, 4, D1 and G2 and tables (Tables 1 and B1) in both the main text and appendices. We will release the ciphertext datasets used in Section 5 to facilitate future security analysis.

## ETHICS STATEMENT

Insecure encryption algorithms can provide users with a false sense of confidence. We caution users against using SELM until a sufficiently secure version is developed and thoroughly validated. We publish our algorithm (and future variants) so weaknesses can be discovered and rectified by the community.

Secure encryption algorithms prevent eavesdropping on private communications. Making such algorithms public so they can be used and analyzed for weaknesses supports the human right to privacy. However, all tools have potential for misuse. Bad actors could use encryption algorithms to avoid law enforcement. This work aims to support privacy and safety, although we acknowledge there is potential for misuse.

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

## APPENDICES

We provide details and experiments omitted in the main text:

1. Appendix A: Detailed review of the Fastfood transform
2. Appendix B: Prompt experiments
3. Appendix C: Hyperparameter sweeps
4. Appendix D: Details on SELM's empirical properties
5. Appendix E: Cryptography details
6. Appendix F: Binary classification model details
7. Appendix G: Distribution-based regularization details
8. Appendix H: Limitations of our work

## A  FASTFOOD TRANSFORM

The Fastfood Transform, as described by Le et al., is a function $f : \mathbb{R}^d \to \mathbb{R}^D$ where $f(x) = Vx$. We remove the scaling matrices and factors from the original Fastfood Transform in the interest of speed. $V$ is a product of diagonal and simple matrices:

$$V = HG\Pi HB \tag{5}$$

$\Pi \in \{0,1\}^{D \times D}$ is a permutation matrix. $H$ is the Walsh-Hadamard matrix, but is computed in practice via the fast Hadamard transform rather than a matrix multiplication. $B$ is a diagonal matrix with random $\{\pm 1\}$ entries on its diagonal. $G$ is a diagonal matrix with random Gaussian entries $\sim \mathcal{N}(0,1)$ on its diagonal. Because of each matrices' structure, they can all be stored in $O(D)$ space and $f(x)$ takes $O(d \log D)$ time, which is significantly improved over $O(nD)$ for both space and time requirements. We note that using the Fastfood Transform for subspace optimization was originally proposed in Li et al. (2018).

## B  PROMPT EXPERIMENTS

Carlini et al. (2022) find three trends that increase LM memorization: (1) larger models, (2) number of times an example has been duplicated and (3) longer contexts (by token count). Inspired by these findings, we vary prompt length and content to improve memorization speeds.

Table B1 shows that a single UUID is the best prompt. We hypothesize that the random nature of a UUID "resets" the LM's pre-trained distribution of likely next tokens.

Table B1: Prompt effect on memorization speed. We memorize 300 tokens of 5 news articles with $d = 3000$. "New Token" adds a new, not pre-trained token to the LM's vocabulary. "Vocab" uses a random token not already present in the message. "Natural Prompt" generates an English prompt for each example: "The first chunk is:", "The second chunk is:", etc. UUID, $2\times$ UUID and $3\times$ UUID are 1, 2 and 3 different UUIDs joined by a "-".

| Prompt | Length | Epochs |
|---|---|---|
| New Token | 1 | $134 \pm 35.1$ |
| Vocab | 1 | $58 \pm 11.0$ |
| Natural Prompt | 4 | $58 \pm 8.4$ |
| UUID | 27 | $50 \pm 7.1$ |
| $2\times$ UUID | 54 | $54 \pm 8.9$ |
| $3\times$ UUID | 76 | $68 \pm 14.8$ |

## C  HYPERPARAMETER SWEEPS

Because deliberate LM memorization in random subspaces has not been explored, we perform an extensive hyperparameter sweep. We aim to minimize the number of epochs before perfect memorization. Table C2 shows the parameters tuned. Note that we do not do a grid search across every possible combination, and that these experiments were performed after choosing UUIDs as the prompt (Appendix B).

Table C2: Hyperparameters and their possible ranges. SAID refers to Structure-aware intrinsic dimension from Aghajanyan et al. (2021).

| Category | Setting | Range of Values |
|---|---|---|
| Intrinsic Dimension | SAID | On/Off |
| | Random Projection | Fastfood projection from $d$ to $D$, Dropout before and/or after Fastfood projection, Tanh, Layernorm, Groupnorm or sigmoid before and/or after Fastfood projection |
| Training | Gradient Clipping | Random log-uniform sampling from $10^{-2}$ to $10^7$ |
| | Optimizer | Adam, AdamW, RAdam, NAdam, AdaFactor, SGD |
| | Learning Rate | Random log-uniform sampling from $10^{-10}$ to 1.0 |
| | LR Schedule | Linear Warmup, Linear Decay, Constant, Reduce on Plateau |
| Regularization | Dropout | $\{0, 0.1, 0.2\}$ |
| | Weight Decay | $\{0, 0.1, 0.2\}$ |
| | $\lambda$ Schedule | Linear Warmup, Constant |

Many standard hyperparameter choices apply to LM memorization in random subspaces: AdamW is the best optimizer and gradient clipping prevent exploding losses, for example.[10] Other hyperparameter choices are specific to memorization in a random subspace. Learning rate needs to be tuned with respect to $d$: we use $2 \cdot 10^{-8}$ for $d = 10,000$ but $1 \cdot 10^{-8}$ for $d = 100,000$. Disabling standard regularizing techniques like dropout and weight decay consistently improves memorization speed. Tuning the learning rate scheduler is difficult because memorization doesn't finish in a fixed number of epochs. If the learning rate drops too low too quickly or if the learning rate stays too high for too long, memorization will take much longer.

We try many different variants on the Fastfood projection. We want to prevent patterns in $\theta^D$ related to the message $m$ from appearing in $\theta^d$ and try various non-linearities on the low-dimensional vector $\theta^d$ and the projected vector $P(\theta^d)$. None of them achieve this security goal nor do they significantly speed up memorization, prompting our turn to distribution-based regularization.

We tune the regularization schedule (Section 5.4) to linearly increase from 0 to the maximum regularization weight over 500 epochs as we find this enables larger $\lambda$ while still successfully memorizing arbitrary data.

## D  SELM EMPIRICAL DISCUSSION

This section contains various in-depth discussions on SELM's empirical properties.

### D.1  ENCRYPTING ARBITRARY DISTRIBUTIONS

Traditional encryption algorithms do not make any assumptions about the inputs; thus, they do not need any proof that any possible input can be encrypted. Because SELM depends on the input data (we minimize loss with respect to $\theta^d$ over the input data), it's not trivial to argue that SELM can encrypt any data. Our experiments demonstrate that more structured inputs are easier to memorize (see Fig. 3). Then, we check if SELM can memorize the least-structured input possible: a uniform distribution over bytes. **SELM successfully memorizes random data, the hardest possible input data.** We find this experiment to strongly support the conclusion that SELM can encrypt any possible input, given sufficient free parameters (a large enough value of $d$).

### D.2  ENCRYPTING LONG MESSAGES

Although SELM cannot encrypt 1,000+ token messages with $d \leq 3,000$ as a complete message (see Fig. 3b, it's still possible to encrypt the different messages. Simply separate the long message $m$ into

---

[10]Note however, that typical gradient clipping L2 norm maxes are between 0.1 and 10; we use a significantly higher value of $10^5$ for random subspace optimization.

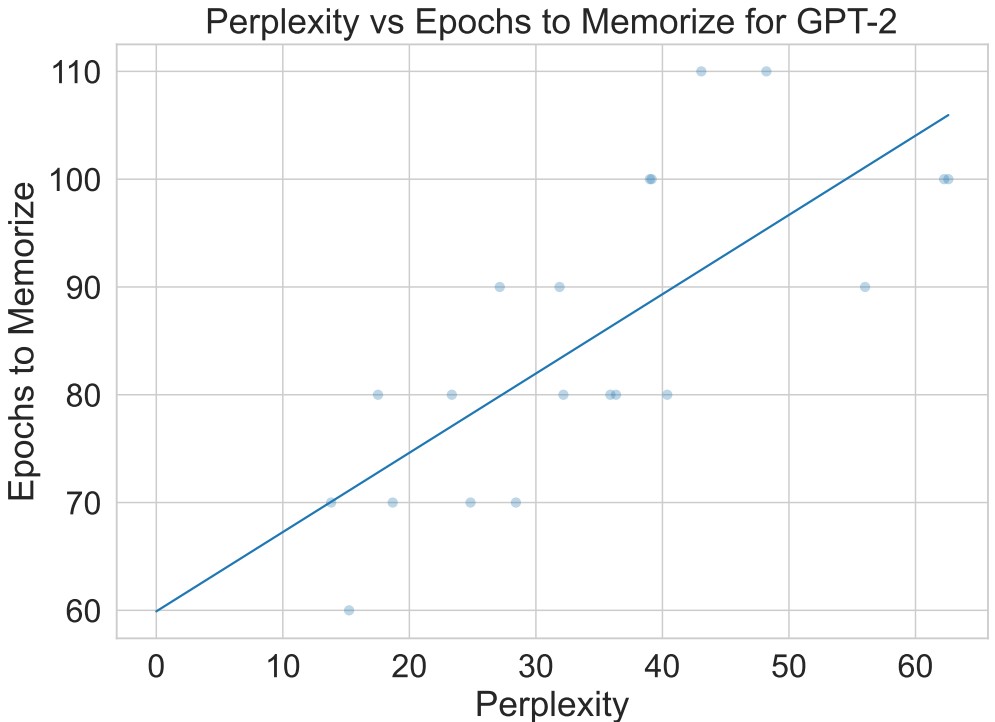

Figure D1: Comparison of message perplexity and epochs needed to memorize for 100-token, non-random messages. Messages with lower initial perplexity are easier to memorize. GPT-2 has a validation perplexity of 29.41 for WikiText-2, 65.85 for Penn Tree Bank, and 37.50 for WikiText-103; perplexities between 10-60 is expected.

$n$ shorter chunks $m_1, m_2, \ldots m_n$. Encrypt each chunk individually, leading to different $c_1, c_2, \ldots c_n$, then send each ciphertext. Through this simple construction, any message of any length can be completely encrypted.

### D.3 PERPLEXITY VS EPOCHS TO MEMORIZE

To support the claim that messages more similar to the pre-training data are easier to memorize, we use message perplexity before memorization to quantify how similar a message is to the pre-training data. We look at 100-token messages from News and PubMed and find an obvious correlation; see Fig. D1. This further supports the idea that messages closer to the original pre-training distribution are easier to memorize.

## E CRYPTOGRAPHY DETAILS

We explain some cryptography concepts in more depth here.

### E.1 USING A DIFFERENT KEY FOR EVERY MESSAGE

A cipher is a pair of functions $\mathcal{E} = (E, D)$ used to encrypt and decrypt data, respectively. Both $E$ and $D$ are parameterized by a key $k$. Given a fixed key $k'$, $D$ is the inverse of $E$: $D(k', E(k', m)) = m$. Deterministic ciphers output the same ciphertext every time for a given key/message pair. *Probabilistic* ciphers can produce one of many ciphertexts for a given key/message pair. A cipher *must* be probabilistic to satisfy IND-CPA; if it is deterministic, Eve could submit two messages $m_0$ and $m_1$ to Alice during Steps 1 and 2 (Section 5.1), then submit the same two messages again during Step 4, and simply compare $c$ with the known ciphertexts of $m_0$ and $m_1$.

---

**Algorithm 1** $E': \mathcal{K}' \times \mathcal{M} \rightarrow (\mathcal{C} \times \mathcal{X})$

---

1: $x \xleftarrow{R} \mathcal{X}$                                                          ▷ Randomly sample $x$ from $\mathcal{X}$
2: $k \leftarrow F(k', x)$
3: $c \leftarrow E(k, m)$
4: **return** $c' = (c, x)$                                              ▷ Return both $c$ and $x$

---

**Algorithm 2** $D': \mathcal{K}' \times (\mathcal{C} \times \mathcal{X}) \rightarrow \mathcal{M}$

---

1: $c, x \leftarrow c'$                                                    ▷ Split $c'$ back into $c$ and $x$
2: $k \leftarrow F(k', x)$                                           ▷ Generate the same key $k$
3: $m \leftarrow E(k, c)$
4: **return** $m$

---

We present an existing construction to turn any deterministic cipher $\mathcal{E} = (E, D)$ into a probabilistic cipher $\mathcal{E}' = (E', D')$ given a pseudo-random function $F$ that takes as input a key $k$ and an integer $x$ and produces a random key $k'$. Rather than use the $k$ to parameterize $E$ and $D$, we use it to parameterize a PRF $F$ and generate a pseudo-random key $k'$ based on a random value $x$ (Algorithm 1). We send $x$ in addition to the ciphertext $c$, which is used to generate the identical pseudo-random key $k'$ when decrypting (Algorithm 2).

This additional step ensures that $E$ uses a different key for every message, making IND-CPA security possible.

Because $F$ is pseudo-random, we can treat $k'$ like it is randomly sampled from $\mathcal{K}$, which is exactly what we do in Section 5.2.

### E.2 IND-CPA vs Modern Cryptanalysis

We use the IND-CPA game (Section 5.1) because it makes no assumptions about the cipher's internal structure. We elaborate more here.

Popular symmetric ciphers like DES and AES are block ciphers: messages, ciphertexts and keys are all fixed-length bit sequences. Webster & Tavares (1985) describes the avalanche criterion, a desired statistical property of block ciphers wherein changing a single bit of the key *or* message should change approximately 50% of the ciphertext bits. Unfortunately, there is no obvious analog for our algorithm.

Linear cryptanalysis (Matsui & Yamagishi, 1992; Matsui, 1994a;b) and differential cryptanalysis (Biham & Shamir, 1993) are two of the most popular cryptanalysis techniques for symmetric block ciphers. Even ignoring that SELM doesn't operate on bit sequences, linear cryptanalysis tries to find linear Boolean equations that consistently hold for many plaintext, ciphertext and key bits. There is simply no analogous structure in SELM that linear cryptanalysis could analyze. Differential cryptanalysis measures statistics of ciphertext bit-differences in plaintext pairs to derive properties about the round keys. SELM doesn't have discrete bit-differences in the ciphertexts; again, analysis designed for block ciphers is not suitable for SELM.

The IND-CPA game, in contrast, makes no assumptions about the message space, the ciphertext space or the internal structures in use. We argue that the IND-CPA game is sufficient analysis for SELM's current incarnation because it successfully finds security flaws.

## F Binary Classification Model Details

We present details for each binary classification model here. Code describing our models will be made available.

### F.1 KNN

We use Scikit-Learn's `KNeighborsClassifier` and do 5-fold cross validation to choose $k$ from 5, 25 or 100.

### F.2   LDA

We use Scikit-Learn's `LinearDiscriminantAnalysis` which fits a Gaussian density to each class and assumes that all classes share the same covariance matrix. We scale the input features to have zero mean and unit variance. By default, the LinearDiscriminantAnalysis classes uses a Ledoit-Wolf lemma for the shrinkage parameter. There are no other hyperparameters.

### F.3   SVM

We use Scikit-Learn's `SVC`. We scale the input features to have zero mean and unit variance. We do a random hyperparameter search using 5-fold cross validation to choose:

1. The kernel function, from a radial basis function, a linear function, a sigmoid function and a polynomial function with degree 3.
2. $\gamma$, the kernel coefficient for the radial basis function kernel, the polynomial kernel and the sigmoid kernel. We sample $\gamma$ from a log-uniform distribution with minimum $10^{-4}$ and maximum $10^{-3}$.
3. $C$, the regularization parameter. We sample $C$ from a log-uniform distribution with minimum $10^{-3}$ and maximum $10.0$.

We evaluate 100 different hyperparameter choices choose the hyperparameters with the highest mean test accuracy across the 5 folds.

### F.4   GRADBOOST

We use Scikit-Learn's `GradientBoostingClassifier` which is a gradient-boosted decision tree model. We use all the default hyperparameter values, the most relevant of which are listed here:

1. We minimize log loss (as in logistic regression).
2. We use a learning rate of 0.1.
3. We use 100 estimators.
4. We fit each learner to all the data.
5. We use a maximum tree depth of 3.

We scale the input features to have zero mean and unit variance.

### F.5   FEED-FORWARD NEURAL NETWORK

We use PyTorch to implement a two-layer feedforward neural network with ReLU non-linearity and dropout after the first activation. We optimize the parameters with AdamW (Loshchilov & Hutter, 2017) with a learning rate of $3 \cdot 10^{-4}$, a weight decay of 0.1, a batch size of 32 and dropout of $p = 0.1$. We train until training loss does not decrease for more than 5 epochs.

When we use the entire ciphertext as the input, our hidden layer has dimension 1,000. When we use the feature function $f(\theta^d)$, our hidden layer has dimension 256.

## G   DISTRIBUTION REGULARIZATION DETAILS

Fig. G2 shows histograms of a ciphertext's values for two ciphertexts from each of the five messages used in Section 5. It's visually apparent that a normal distribution is a natural fit for a target distribution.

To measure the Wasserstein distance between a ciphertext $c$ and an $d$-dimensional sample $x$ from a normal distribution $\mathcal{N}(0, \sigma^2)$, we sum up the area between $c$'s empirical CDF and $\mathcal{N}$'s theoretial CDF. Algorithm 1 demonstrates the Python code to calculate the error term. We use the trapezoid rule to calculate an approximate integral, but in practice the $dx$ is less than $10^{-6}$, so it is a precise approximation. We note that sorting the values in $c$ to construct the empirical CDF likely harms optimization.

## H   LIMITATIONS

Our proposed algorithm currently has three noteworthy limitations: speed, hardware requirements, and security.

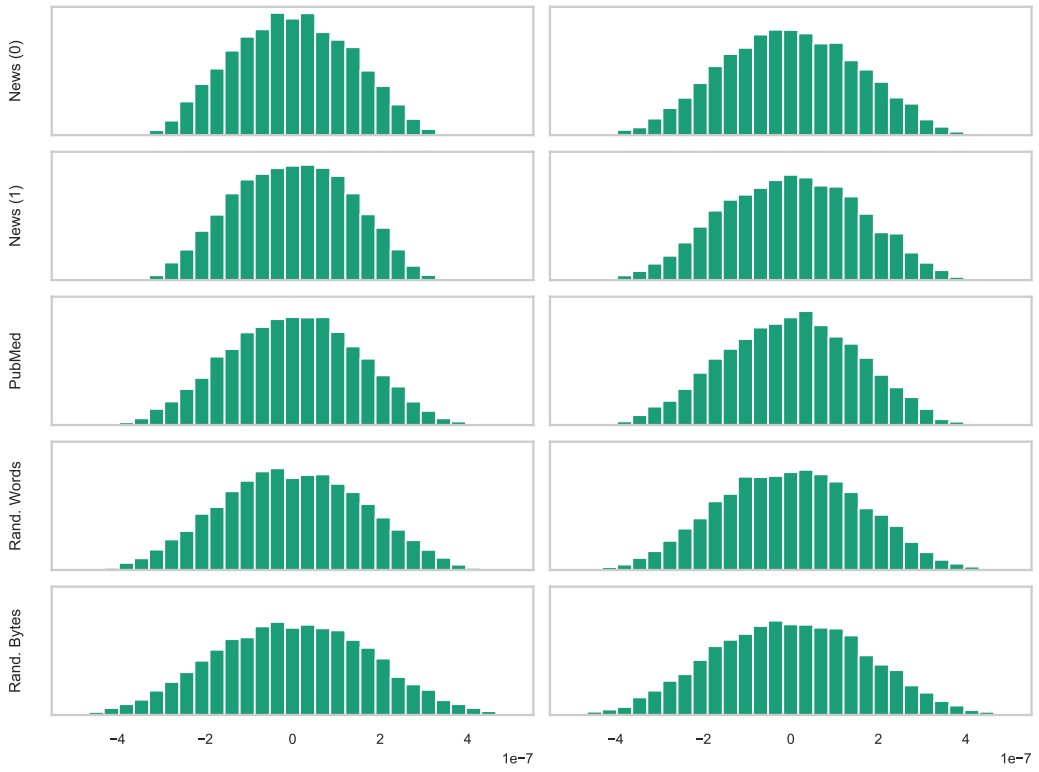

Figure G2: Histograms of the values in two ciphertexts $c$ each for the five files used in Section 5. Even without regularization, ciphertexts have approximately normal distributions.

**Listing 1** Python code to measure the 1D Wasserstein distance between $c$ and a $d$-dimensional sample of $\mathcal{N}(0, \sigma^2)$.

```python
import numpy as np
import scipy.stats

def error(c, sigma):
    """
    c: list of ciphertext values
    sigma: standard deviation
    """
    n = len(c)
    ordered = sorted(c)

    ecdf = np.arange(0, n) / n
    cdf = scipy.stats.norm(0, sigma).cdf(ordered)
    return np.trapz(np.abs(edcf - cdf), ordered)
```

Our algorithm is orders of magnitude slower than typical modern encryption algorithms because of its limited hardware support and higher complexity. However, today's encryption algorithms often have hardware support (e.g., specialized CPU instructions for AES). We expect hardware and software support for large language models and neural networks in general to narrow the gap.

The required hardware (GPUs) limits general use of our algorithm. Again, we expect that as neural networks become more and more integrated in everyday use, this limitation will ease over time.

Our security analysis is limited due, in part, to our algorithm's novelty. Standard cryptanalyses are not well-suited to our algorithm because of the assumptions they make about ciphers' internal structures.

We hope that our work inspires specialized cryptanalysis for neural network-based cryptography algorithms.

More broadly, our work only investigates one LM (GPT-2$_{small}$) as SELM's backbone due to limited computing resources. Investigating other autoregressive LMs of different sizes and architectures would help us better understand LM memorization capabilities and SELM.

We also only evaluate security with 100-token messages and ciphertexts with $d = 10K$. More comprehensive explorations of message lengths and ciphertext dimensions would strengthen our conclusions about SELM's security properties.

Table H3: Performance comparisons with widely-used symmetric encryption algorithms.

| Algorithm ($d$) | Length (Tokens) | Bytes/Sec | $\times$ Larger |
|---|---|---|---|
| AES 256 | - | 640M | 1.0 |
| TripleDES | - | 31M | 1.0 |
| Camellia | - | 150M | 1.0 |
| Cast5 | - | 100M | 1.0 |
| SELM ($10^3$) | 100 | 6.8 | 8.8 |
| SELM ($10^4$) | 100 | 26.4 | 87.6 |
| SELM ($10^5$) | 100 | 31.2 | 875 |
| SELM ($10^4$) | 1000 | 50.3 | 8.7 |
| SELM ($10^5$) | 1000 | 113 | 89.1 |

