# OpenReview forum: "Memorization for Good: Encryption with Autoregressive Language Models"
_ICLR.cc/2024/Conference — ICLR 2024 Conference Withdrawn Submission_

### Official Review · Reviewer_dUZZ · 2023-10-31

**Soundness:** 2 fair
**Presentation:** 2 fair
**Contribution:** 2 fair
**Rating:** 3
**Confidence:** 5

**Summary:**

This paper proposes a new secret-key encryption scheme based on the idea that LLMs memorize. At a high level, in this approach Alice and Bob both have access to a public LLM but Alice, who wants to send message m to Bob, will find a fine-tuning to the LLM that is based on the secret key k and will allow Bob to extract the planted message m from the fine-tuned model. The idea is that the changes to the model are hidden behind a random projection that is based on the secret key and hence the eavesdropper, at least through an obvious attempt, can not find the information that Bob is able to find and extract the message m.

The paper also does experimental attacks, trying to break its own scheme, and their best efforts *does* indeed break the proposed scheme. The author(s) plan ideas for future improvements that might resist the attacks.

**Strengths:**

Aiming to find a new encryption scheme.

**Weaknesses:**

I do not think the paper is looking for the right application for LLM's memorization. Encryption is a very subtle task and has its own key parameters, such as the size of the ciphertext and the ability to resists attacks.

This paper's proposal leads to quite large ciphertexts, and even a "black-box" attack based on ML (itself) is breaking the proposed scheme. So I don't see why this proposal might actually lead to a useful scheme.

I also believe a new proposal for encryption should be submitted to a cryptography venue to get proper checks, not a learning venue. And in doing so, the presentation should be fully accessible to crypto audience, not written the way it is, with fully clear exposition of the assumptions, tools, notions, etc, so that the crypto community can fully understand and review the new proposal for encryption.

I would be more lenient if the proposal was a public key scheme, for which the crypto community is much more interested in new proposals. But to me, the bar for proposing a new secret key scheme is way higher.

**Questions:**

You say in page 6: "Other symmetric ciphers like the Data Encryption Standard (National Bureau of Standards, 1977, DES) and the Advanced Encryption Standard (Pub, 1999, AES) do not have this flexibility; the ciphertext is always the same size as the message."
How come this is a weakness? Cannot you always artificially *increase* their ciphertext, if that is needed as a "feature"?

Your security game in page 6 is not clear. What does "encrypt them" in step 2 of your re-framed game is done? which one of m_0 or m_1 is encrypted?

in page 7 and 8, you say that "operating on bit sequences" is a factor that *limits* previous crypto-analysis approaches. But is not it that without loss of generality? As far as I understand, everything can be re-written in zeros and ones.

---

> ### Author Response · Authors · 2023-11-22
>
> We thank the reviewer for their time and effort spent reading and engaging with our work. We’d like to answer your questions as best we can.
>
> **You say in page 6: "Other symmetric ciphers like the Data Encryption Standard (National Bureau of Standards, 1977, DES) and the Advanced Encryption Standard (Pub, 1999, AES) do not have this flexibility; the ciphertext is always the same size as the message." How come this is a weakness? Cannot you always artificially increase their ciphertext, if that is needed as a "feature"?**
> Good point; you are correct.
>
> **Your security game in page 6 is not clear. What does "encrypt them" in step 2 of your re-framed game is done? which one of m_0 or m_1 is encrypted?**
> Encrypt both $m_0$ and $m_1$ to form a set of examples ${(c_i, m_i)}^t_{i \in \{0, 1\}}$. Then the training set is a set of ciphertexts with binary labels, 0 or 1, indicating which message produced the ciphertext. Since encryption is not deterministic, the ciphertexts will all be different.
>
> We apologize for the typo and will fix it.
>
> **in page 7 and 8, you say that "operating on bit sequences" is a factor that limits previous crypto-analysis approaches. But is not it that without loss of generality? As far as I understand, everything can be re-written in zeros and ones.**
> You are again correct. Because existing encryption algorithms operate on bit sequences, existing cryptanalyses also look for bit-level patterns. We could apply them to our ciphertexts, but we think the lack of bit-level operations would make these analyses unlikely to work. We will correct this language in future versions.

---

### Official Review · Reviewer_vHhk · 2023-11-04

**Soundness:** 2 fair
**Presentation:** 1 poor
**Contribution:** 3 good
**Rating:** 3
**Confidence:** 4

**Summary:**

In this paper, the authors proposed an LM-based symmetric key scheme, and they claimed that autoregressive LM can encrypt arbitrary data as a real vector and recover it.

**Strengths:**

I agreed that it is possible to perform encryption and decryption based on LM, and this paper attempted to construct a corresponding method.

**Weaknesses:**

There must be more exact security proof or argument for the proposed "symmetric key encryption" method. Although the proposed method is based on a random project, Pk is just linear transformations such as a combination of orthogonal matrices and diagonal matrices. The activation function may act as a non-linear transformation, but this is unclear from the algorithm description. Since this structure is similar to the primary design principles of existing block ciphers, conventional cryptanalysis methods may also be applied by adapting to the domain. However, this has not yet been attempted in this paper. Instead, the IND-CPA game was modified to fit the classification problem. Nevertheless, looking at the presented simulation results, such as Fig. 4, the distinguishability between two distributions is far from negligible in all three cases. Therefore, it seems to be evidence that it is not secure.

**Questions:**

It seems that the secret key k is used for the generation of projection P_k, but what is the specific generation method of projection from the given k? What is the size of different subspaces generated from k?
Are projections always orthogonal to each other? If not, is there any possibility of information leaking from subspaces close to each other?

**Details Of Ethics Concerns:**

I have no concerns

---

> ### Author Response · Authors · 2023-11-22
>
> We thank the reviewer for their time and effort spent reading and engaging with our work. We’d like to answer your questions as best we can.
>
> **It seems that the secret key k is used for the generation of projection P_k, but what is the specific generation method of projection from the given k?**
> We use torch.randn, torch.randperm and torch.randint in combination with torch.set_seed to generate the required matrices for a FastFood transform. In a truly secure implementation, you should use cryptographically secure methods of generating random numbers.
>
> **What is the size of different subspaces generated from k?**
> The subspace size is the intrinsic dimension $d$, which is a hyperparameter. We discuss the effects of $d$ in Section 4.3
>
> **Are projections always orthogonal to each other?**
> No.
>
> **If not, is there any possibility of information leaking from subspaces close to each other?**
> We don’t think so, because there’s no way an attacker could know that two subspaces are close to each other.

---

### Official Review · Reviewer_wc1D · 2023-11-04

**Soundness:** 2 fair
**Presentation:** 2 fair
**Contribution:** 2 fair
**Rating:** 3
**Confidence:** 3

**Summary:**

The paper proposes an 'encryption' scheme that uses language models. The basic idea is as follows:

1. The two parties share the public parameters of the model.
2. Alice finetunes the changes in the model's parameters and sends them across to Bob.
3. Bob uses the changes to recover the original message.

This doesn't work as is, but the others propose using a hidden map that maps from a low-dimensional space to the full space of the network. The hidden map needs to be known to both Alice and Bob.

**Strengths:**

Interesting model but several drawbacks.

**Weaknesses:**

There is not much in the way of security, which is the most essential part of any encryption/decryption mechanism.

**Questions:**

Can you provide evidence for why the scheme could be secure? Traditionally, the standard schemes have concrete quantitative assumptions under which one can prove security.

---

> ### Author Response · Authors · 2023-11-22
>
> We thank the reviewer for their time and effort spent reading and engaging with our work. We’d like to answer your questions as best we can.
>
> **Can you provide evidence for why the scheme could be secure? Traditionally, the standard schemes have concrete quantitative assumptions under which one can prove security.**
> We think that we explain the intuition for why SELM is secure in Section 3 (in the paragraph starting “Intuitively…”) and we also provide extensive quantitative experiments in Section 5 (titled Security).
>
> Furthermore, symmetric encryption algorithms are almost never provably secure. AES, DES and other symmetric encryption algorithms **are not provably secure.**

---

### Official Review · Reviewer_XtDw · 2023-11-04

**Soundness:** 4 excellent
**Presentation:** 4 excellent
**Contribution:** 3 good
**Rating:** 6
**Confidence:** 3

**Summary:**

This paper introduced a novel approach that leverages the (unintended) memorization in LLMs to create a cryptographic system named SELM. They demonstrate for the first time that LLMs can be harnessed to implement a system capable of lossless encryption and decryption. I believe this work can potentially catapult a new line of research.

**Strengths:**

1. A novel approach to harness unintended memorization in LLM for the first time.

2. The overhead of the proposed cryptographic system does not depend on the absolute number of trainable parameters in LLMs but rather on a smaller subset. This reduces the communication bandwidth requirements, a crucial bottleneck for interactive crypto-systems.


3. The authors have presented a rigorous analysis of  (intuitive) security provided by the proposed crypto-system.

4. Authors have also shown how the security of SELM can be improved using various regularization techniques.

**Weaknesses:**

1. The proposed protocol does not provide a provable security guarantee.


2. Only evaluates short message lengths (<1000 tokens), so scalability is not yet proven.

3. Lack of discussion on how the proposed method stacks up against traditional crypto methods like MPC (such as extended oblivious transfer or VOLE) regarding communication and computation requirements. In particular, a discussion on the tradeoffs between message length, ciphertext size, and compute and communication requirements.


**In summary, while the contributions are novel, the scalability of the proposed crypto-system is questionable, especially as an alternative to the existing crypto-primitives such as homomorphic encryption and MPC**.  Moreover, the challenge of transmitting messages securely is not new and is largely considered resolved, especially when contrasted with the complexities of performing computations on encrypted data (e.g, using homomorphic encryption).  I'm open to increasing the score, provided critical concerns are addressed.

**Questions:**

1. Can we further improve memorization speed using Prompt engineering strategies?


 2. What attack vectors or cryptanalyses are the biggest threats to SELM security?


3. Can we employ off-the-shelf LLM optimization techniques to reduce the computation burden of the SELM without compromising its security guarantee?

4. What are the key (potential) backdoors in LLM to undermine the security of SELM?

---

> ### Author Response · Authors · 2023-11-22
>
> We thank the reviewer for their time and effort spent reading and engaging with our work. We’d like to answer your questions as best we can.
>
> **Can we further improve memorization speed using Prompt engineering strategies?**
> Most likely, yes. We perform some experiments in Appendix B on this and find that the UUID prefix leads to the best memorization speed; we think this is because it “resets” the LM context.
>
> **What attack vectors or cryptanalyses are the biggest threats to SELM security?**
> In our experiments, gradient-boosted decision trees do the best on the security game. However, we’re aware that our security experiments are not as rigorous as the decades of attacks on existing algorithms like AES and we expect there to be more attacks that are stronger against SELM.
>
> **Can we employ off-the-shelf LLM optimization techniques to reduce the computation burden of the SELM without compromising its security guarantee?**
> Yes! We think that FlashAttention is an example of an LM optimization that was released during this work and would likely lead to improved wall clock memorization speed. This is part of why we’re excited about SELM: we can take advantage of LM optimizations and apply them for free to SELM.
>
> **What are the key (potential) backdoors in LLM to undermine the security of SELM?**
> We’re not aware of any backdoors in LLMs that would undermine the security of SELM, but we agree that it’s an interesting area for further work.